Effects of stigmata maydis on the methicillin resistant Staphylococus aureus biofilm formation

Shang Fei
Li Long
Yu Lumin
Ni Jingtian
Chen Xiaolin
Xue Ting xuet@ahau.edu.cn
School of Life Sciences, Anhui Agricultural University , Hefei, Anhui , China
Tulkens Paul
Electronic publication date: 2019 Feb 26
Publication date: 2019
Volume: 7
Electronic Location ID: e6461
Received 2018 Aug 31; Accepted 2019 Jan 15
Copyright: © 2019 Shang et al.
Copyright year: 2019
Copyright holder: Shang et al.
License: This is an open access article distributed under the terms of the Creative Commons Attribution License, which permits unrestricted use, distribution, reproduction and adaptation in any medium and for any purpose provided that it is properly attributed. For attribution, the original author(s), title, publication source (PeerJ) and either DOI or URL of the article must be cited.
License URL: https://creativecommons.org/licenses/by/4.0/

Keywords: MRSA, Bovine mastitis, Biofilm, Vancomycin, Stigmata maydis

Funding: National Natural Science Foundation of China 31672571 and 31371324 This work was supported by the National Natural Science Foundation of China (No. 31672571 and No. 31371324). The funders had no role in study design, data collection and analysis, decision to publish, or preparation of the manuscript.

==============================
Background

Mastitis is an inflammatory reaction of the mammary gland tissue, which causes huge losses to dairy farms throughout the world. Staphylococcus aureus is the most frequent agent associated with this disease. Staphylococcus aureus isolates, which have the ability to form biofilms, usually lead to chronic mastitis in dairy cows. Moreover, methicillin resistance of the bacteria further complicates the treatment of this disease. Stigmata maydis (corn silk), a traditional Chinese medicine, possess many biological activities.

Methods

In this study, we performed antibacterial activity assays, biofilm formation assays and real-time reverse transcription PCR experiments to investigate the effect of stigmata maydis (corn silk) on biofilm formation and vancomycin susceptibility of methicillin-resistant Staphylococcus aureus (MRSA) strains isolated from dairy cows with mastitis.

Results

In this study, the aqueous extracts of stigmata maydis inhibited the biofilm formation ability of MRSA strains and increased the vancomycin susceptibility of the strains under biofilm-cultured conditions.

Conclusion

This study proves that the aqueous extracts of stigmata maydis inhibit the biofilm formation ability of MRSA strains and increase the vancomycin susceptibility of the MRSA strains under biofilm-cultured conditions.

Introduction

Staphylococcus aureus is a major pathogen that can cause a series of infections in both hospital and community environments (Archer & Climo, 2001; Lowy, 1998). The infections caused by this bacterium are complicated by frequent and multiple antibiotic use in medical treatment in the past several decades (Lowy, 2003; Queck et al., 2009). Methicillin-resistant Staphylococcus aureus (MRSA) arose in the 1960s, after methicillin became the antibiotic of first choice for Staphylococcus aureus infections because of the wide spread of penicillin-resistant strains (Richmond, 1979). In the first few years, MRSA strains only affected people who were associated with risk factors, such as surgery, recent admittance, or long-term residence in care facilities. However, community-associated MRSA affections are now prevalent in the general population and pose a serious threat to public health worldwide (Chambers, 2001; Hunt et al., 1999; Hiramatsu et al., 2001; Naimi et al., 2001). In addition, the treatment for these infections becomes more difficult and complicated due to the development of biofilms (Kiedrowski & Horswill, 2011).

The formation of a biofilm is characterized by the structure of a population of bacteria encased within a self-produced extracellular matrix of exopolysaccharide, proteins and some micromolecules, such as DNA (O’Gara, 2007). It is well known that the properties of biofilm populations are largely different from planktonic cell populations, and these contribute to better adaptation to the host environment. The presence of glycocalyx layers protects the enclosed bacteria from host defenses and resists the access of antibiotics (Atshan et al., 2012b; Fedtke, Götz & Peschel, 2004; Joh et al., 1999). It has been reported that biofilms can resist antibiotic concentration 10–10,000-fold higher than those required to inhibit the growth of their planktonic counterparts (Atshan et al., 2015; Jefferson, Goldmann & Pier, 2005). Indeed, the ability of biofilm formation in MRSA can lead to resistance to most currently used antibiotics (Ando et al., 2004). Therefore, biofilm formation brings great challenges for the infection treatment, eventually leading to chronic infections, which can be difficult to eradicate (Kiedrowski & Horswill, 2011; Petrelli et al., 2008; Pozo & Patel, 2007).

Bovine mastitis is a disease causing substantial economic loss in the dairy industry worldwide (Hillerton & Berry, 2005; Huijps, Lam & Hogeveen, 2008; Szweda et al., 2014). Although many species of etiological microorganisms have been isolated from bovine mastitis (Watts, 1988), Staphylococcus aureus is frequently the cause for the majority of this economic loss (Kozytska et al., 2010; Malinowski & Kłossowska, 2010; Piepers et al., 2007). Since Staphylococcus aureus has the ability to form biofilms and is resistant to many antibiotics, it causes chronic bovine mastitis, which is difficult to treat (Cramton et al., 1999). Moreover, methicillin resistance of Staphylococcus aureus could further complicate the treatment of this disease (Joshi et al., 2018; Lowy, 2003).

Many plants have been used as a traditional Chinese medicine for the treatment of various diseases in China. The medicinal value of plants lies in some constituents that have definite biological functions. In recent years, many Chinese medicines have been reported to have antimicrobial effects.

Stigmata maydis (corn silk) refers to the stigmas of the female flowers of maize, which contain proteins, carbohydrates, vitamins, Ca, K, Mg and Na salts, fixed and volatile oils, steroids, such as sitosterol and stigmasterol, alkaloids, saponins, tannins, and flavonoids (Bhaigyabati et al., 2011; Hasanudin, Hashim & Mustafa, 2012). Many biological activities of corn silk constituents have been reported. Extracts of corn silk inhibited tumour-necrosis factors (TNF) and lipopolysaccharide (LPS)-induced cell adhesion, but not cytotoxic activity or TNF production (Habtemariam, 1998). Moreover, volatiles from corn silk showed antifungal activity (Zeringue, 2000). In addition, extracts of corn silk displayed antioxidant activity on the level of lipid peroxidation (Bhaigyabati et al., 2011). Corn silk has also been used as a remedy for acute inflammation of the urinary system, such as urethritis, cystitis and prostatitis in many parts of the world, and it has also been used as an oral antidiabetic agent in China for decades (Hasanudin, Hashim & Mustafa, 2012). However, whether stigmata maydis is associated with biofilm formation and antibiotic resistance in bacteria has not been reported.

In this study, we aim to investigate the effect of stigmata maydis aqueous extracts on growth and biofilm formation of MRSA strains isolated from dairy cows with mastitis. The new finding in this study may provide new clues or potential methods to the efficient antibiotic treatment of this disease.

Materials and Methods

Bacterial strain and growth condition

Staphylococcus aureus MRSA strains SA2 and SA3 used in this study were isolated from dairy cows with mastitis. The two MRSA strains are mecA positive and susceptible to vancomycin (vancomycin minimum inhibitory concentration (MIC) SA2 0.5 mg/L; SA3: one mg/L), and sensitive to chloromycetin, and resistant to ampicillin, erythromycin and oxacillin. The strains were grown at 37 °C in tryptic soy broth (TSB) containing 0.25% glucose media (Oxoid, Basingstock, UK).

Aqueous extraction of stigmata maydis

One g of stigmata maydis powder was suspended in 10 mL water and soaked for 24 h. The supernatant was dried by vacuum freeze drying and then supernatants were mixed into a 100 mg/mL extract.

Antimicrobial activity assay

The method was performed as described previously as follows (Chen et al., 2015). Colonies of MRSA strains were picked into two mL of TSB medium and cultivated at 37 °C with shaking at 200 rpm for 16 h. Then the overnight cultures were inoculated into fresh TSB medium and this was diluted to a final optical density (600 nm) of 0.05, which was dispensed into 96-well plates (Corning, NY, USA) containing serial dilutions of aqueous extracts of stigmata maydis (with appropriate vancomycin, if needed). Plates were incubated at 37 °C for 12 h and then 10-fold serial dilutions of cultures were performed by successive transfer (0.1 mL) through seven microfuge tubes containing 0.9 mL of TSB. The 100 μL dilutions were dropped onto LB agar plates and viable colonies were counted via their colony-forming units (CFU) on TSB agar plates after incubation at 37 °C for 24 h. The survival rate of the control group without exposure to Sanguisorba officinalis L. was designated as 100%. CFU of the test groups were all compared with that of the control group. Experiments were repeated three times with four parallels. Experiments were repeated three times with four parallels.

Biofilm assays

The method for biofilm quantification was performed as described previously and modified as follows (Chen et al., 2015; Xue, Chen & Shang, 2014). MRSA strains were grown in TSB (containing 0.5% glucose) for 16 h and diluted 1:100 into fresh TSB. The diluted cultures were transferred into sterile 96 well flat-bottomed tissue culture plates and incubated at 37 °C for 24 h. Aqueous extracts of stigmata maydis were added to the TSB media with diluted cultures at different concentrations. The adherent bacteria were stained with crystal violet, and the excess stain was washed off gently with slowly running water. The biomass of the biofilm was determined using a MicroELISA auto-reader (Bio-Rad Co., Hercules, CA, USA) at a wavelength of 560 nm under single-wavelength mode (Pozzi et al., 2012; Ziebuhr et al., 1997).

Total RNA isolation, cDNA generation and real-time PCR processing

Overnight cultures of MRSA strains were diluted 1:100 in TSB medium and the aqueous extracts of stigmata maydis was added to the experimental group at different concentration (0.5, one and two mg/mL, respectively). The method used for the total RNA isolation of the treated group was the same as that of the non-treated group. The cells were grown to the late exponential phase in 24-well plates (Corning, NY, USA). Subsequently, they were collected and resuspended in Tris–EDTA buffer (pH 8.0) containing 10 g/L of lysozyme and 40 mg/L of lysostaphin. After incubation at 37 °C for 5 min, cells were prepared for total RNA extraction using the Trizol method (Invitrogen, Carlsbad, CA, USA), and residual DNA was removed with DNase (RNase free; TaKaRa, Kusatsu, Shiga Prefecture, Japan). RT real-time PCR was performed with a PrimeScript 1st Strand cDNA synthesis kit and SYBR Premix Ex Taq (TaKaRa, Kusatsu, Shiga Prefecture, Japan) using a StepOne real-time PCR system (Applied Biosystems, Foster City, CA, USA). The quantity of cDNA measured by real-time PCR was normalized to the abundance of 16S cDNA (Chen et al., 2000). All real-time reverse transcription PCR (RT-PCR) assays were repeated at least three times with similar results. The primers used in this study were listed in Table 1.

Table 1 Oligonucleotide primers used in this study.

Primer name	Oligonucleotide (5′-3′)	
rt-icaA-f	TTTCGGGTGTCTTCACTCTAT	
rt-icaA-r	CGTAGTAATACTTCGTGTCCC	
rt-icaB-f	CCTATCCTTATGGCTTGATGA	
rt-icaB-r	CATTGGAGTTCGGAGTGA	
rt-icaC-f	TACTGACAACCTTGAATTACCA	
rt-icaC-r	AATAGCCATACCATTGACCTAA	
rt-icaD-f	CCAGACAGAGGGAATACC	
rt-icaD-r	AAGACACAAGATATAGCGATAAG	
rt-icaR-f	TTATCTAATACGCCTGAGGAAT	
rt-icaR-r	GGATGCTTTCAAATACCAACT	

Statistical analysis

The data was analyzed using statistical software SPSS by a one-way ANOVA method, the test results were (mean ± standard deviation). The paired t-test was used for statistical comparisons between groups. The level of statistical significance was set at a P-value of ≤0.01.

Results

Effects of stigmata maydis on the growth curve of MRSA strain

The growth rates of the cells were tested when they were grown in TSB medium with different concentrations of aqueous extracts of stigmata maydis. The results showed that the growth rates of the bacteria did not change when the external concentration of stigmata maydis was 2.5, 5 or 10 mg/mL, but when the concentration of stigmata maydis was 25 mg/mL, the growth of the bacteria was a little inhibited (Fig. 1A). These data indicate that aqueous extracts of stigmata maydis did not affect the growth curves of the MRSA strains.

Figure 1 Effect of stigmata maydis aqueous extract on growth of MRSA strains.

(A) The growth curves of MRSA strains SA2 and SA3 cultured in tryptic soy broth medium with or without specific concentrations of stigmata maydis extract. The results represent a mean of three independent experiments. (B) Colony-forming unit assays of MRSA strains SA2 and SA3. Colony counts of strains SA2 and SA3 were compared after 12 h of incubation at 37 °C with or without addition of stigmata maydis. The colony counts of the test group cultured with different concentrations of stigmata maydis were all compared with that of the control group (without stigmata maydis), the survival rate of which was designated as 100%.

Furthermore, in order to examine the antibacterial activity of the aqueous extracts of stigmata maydis against Staphylococcus aureus in vitro, antibacterial assays were performed. After exposure to extract of stigmata maydis at different concentrations for 12 h at 37 °C, the cells of MRSA strains were inoculated into fresh TSB and then spread onto the TSB agar plates. After cultivating for 24 h at 37 °C, the colony forming units of the bacteria were counted and compared. As is shown in Fig. 1B, the survival rate of the control group without exposure to stigmata maydis was designated as 100%. With the increase of the concentration of stigmata maydis, the survival rates of the MRSA strain SA2 and SA3 also did not change significantly. These data confirmed that stigmata maydis does not have antibacterial activity against the MRSA strains.

Effects of stigmata maydis on the biofilm formation of MRSA strain

To examine whether an aqueous extract of stigmata maydis would affect the biofilm formation of Staphylococcus aureus, we performed biofilm assays. As shown in Fig. 2A, strains of the control group without aqueous extract of stigmata maydis formed obvious biofilms, when stigmata maydis at different concentration was added, biofilm formation of the bacteria significantly decreased. When the concentration of stigmata maydis reached two mg/mL, no biofilm was observed. In addition, the quantity of biofilm formation was further tested using a MicroELISA autoreader. We found that biofilm quantity decreased with the increase of stigmata maydis concentration (Fig. 2B).

Figure 2 Effect of stigmata maydis aqueous extract on biofilm formation of MRSA strains.

The cells of strains SA2 and SA3 were cultured in 96-well plates for 24 h at 37 °C, and the tigmata maydis extract was added in the tryptic soy broth at concentrations of zero, 0.5, one and two mg/mL, respectively. (A) Photographs of the 96-well plates were taken after staining with crystal violet. (B) The biomass that adhered to the plate after staining with crystal violet was measured by a MicroELISA auto-reader at a wavelength of 560 nm. The results represent a mean of three independent experiments. (** represents P < 0.01).

Effects of stigmata maydis on the biofilm genes expression

The transcript levels of biofilm-associated genes were determined by performing real-time RT-PCR experiments. As is shown in Fig. 3, the transcript levels of icaA, icaB, icaC and icaD were significantly decreased upon the addition of stigmata maydis. Moreover, with increased of stigmata maydis concentrations, the inhibitory effect of stigmata maydis on the ica operon was stronger, indicating that stigmata maydis affected the transcription of the ica operon in a concentration-dependent manner. To further investigate how stigmata maydis regulates the ica operon, we examined the transcript level of icaR, which has been identified as the repressor of the ica operon. Results showed that the transcript level of icaR increased by adding stigmata maydis to the culture medium with Staphylococcus aureus, confirming that stigmata maydis influences the ica operon through the transcriptional regulator icaR.

Figure 3 Comparison of the relative transcript levels of several biofilm-associated genes.

The transcript levels of icaA, icaD, icaB, icaC and icaR were measured by performing real-time reverse transcription-PCR in strains SA2 (A) and SA3 (B). The stigmata maydis extract was added to the culture medium at concentrations of zero, 0.5, one and two mg/mL, respectively. (** represents P < 0.01).

Effects of stigmata maydis on the vacomycin susceptibility in the planktonic MRSA growth

To examine the effect of stigmata maydis on vacomycin susceptibility of the MRSA strains, the antibacterial assays were performed in the planktonic cultured MRSA strains. As is shown in Fig. 4A, in the presence of a low concentration of vancomycin (1/4 MIC concentration), with increased concentrations of stigmata maydis, the survival rates of MRSA strain SA2 and SA3 did not apparently change. As is shown in Fig. 4B, similar results were also observed in the planktonic cultured MRSA strains in the presence of a low concentration of vancomycin (1/2 MIC concentration). These data confirmed that stigmata maydis does not affect the vacomycin susceptibility of the planktonic cultured MRSA strains.

Figure 4 Colony-forming unit assays of the planktonic-cultured MRSA strains SA2 and SA3 in the presence of vancomycin.

The colony counts of the test group cultured with different concentrations of stigmata maydis were all compared with that of the control group (without stigmata maydis), the survival rate of which was designated as 100%. (A) MRSA strains were cultured with 1/4 MIC concentration of vancomycin (SA2: 0.125 μg/mL, SA3: 0.25 μg/mL). (B) MRSA strains were cultured with 1/2 MIC concentration of vancomycin (SA2: 0.25 μg/mL, SA3: 0.5 μg/mL).

Effects of stigmata maydis on the vacomycin suspectibility in the biofilm MRSA growth

However, the results of the antibacterial assays performed in the biofilm-condition cultured MRSA strains were different with those performed in the planktonic-cultured MRSA strains. As shown in Fig. 5, the survival rates of MRSA strain SA2 and SA3 were all decreased with the increased concentrations of stigmata maydis. When the two mg/mL stigmata maydis concentration was added, the survival rates of strains SA2 and SA3 were decreased to about 50% in the presence of 1/4 MIC concentration vancomycin (Fig. 5A), and the survival rates of strains SA2 and SA3 were decreased to about 30% in the presence of 1/2 MIC concentration vancomycin (Fig. 5B). These results confirmed that, in the biofilm-cultured condition, the aqueous extract of stigmata maydis could enhance the vancomycin susceptibility of the MRSA strains.

Figure 5 Colony-forming unit assays of the biofilm-condition cultured MRSA strains SA2 and SA3 in the presence of vancomycin.

The colony counts of the test group cultured with different concentrations of stigmata maydis were all compared with that of the control group (without stigmata maydis), the survival rate of which was designated as 100%. (A) MRSA strains were cultured with 1/4 MIC concentration of vancomycin (SA2: 0.125 μg/mL, SA3: 0.25 μg/mL). (B) MRSA strains were cultured with 1/2 MIC concentration of vancomycin (SA2: 0.25 μg/mL, SA3: 0.5 μg/mL). (** represents P < 0.01).

Discussion

The data in this study showed that stigmata maydis aqueous extract did not affect the growth of MRSA strains SA2 and SA3, and had no apparent antibacterial activity against these strains; however, it significantly inhibited the biofilm formation of these strains only at a low concentration. This is consistent with the previous findings by Lin et al. (2011) reported inhibitory effects of 1,2,3,4,6-Penta-O-galloyl-D-glucopyranose (an active ingredient in plants) on biofilm formation by Staphylococcus aureus. Moreover, some kinds of Chinese medicine were reported that inhibited the growth and biofilm formation of Staphylococcus aureus (Chen et al., 2015; Fan et al., 2014).

According to previous work, biofilm formation in Staphylococcus aureus includes two steps: attachment to the material surface and then the formation of microcolonies and multilayered cell clusters surrounded by a slimy matrix, which has been characterized as polysaccharide intercellular adhesin (PIA). PIA is produced by the enzymes coded in an operon composed of four open reading frames icaA, icaD, icaB and icaC. The attachment process and accumulation of bacteria is associated with several adhesion genes such as fnbpA, fnbpB (encoding fibronectin binding proteins A and B), fib (encoding fibrinogen binding protein), clfA (encoding clumping factors A), clfB (encoding clumping factor B), aap (accumulation-associated protein), ssp1 (staphylococcal surface protein), atlE (major autolysin) and bap (biofilm-associated protein), etc. (Atshan et al., 2012a). In this study, we tested the transcript levels of these genes by performing real-time RT-PCR experiments. The results showed that only the transcription of ica operon and its regulatory gene icaR changed with the addition of stigmata maydis. The transcript levels of the adhesion genes exhibited no apparent change. A previous study also reported the similar results indicating that lipoteichoic acid inhibited Staphylococcus aureus biofilm formation through inhibiting ica gene expression, but not through the adhesive matrix molecules (MSCRAMMs) genes, such as clfA, clfB, cna (encoding collagen binding protein), and eno (encoding laminin binding protein) (Ahn et al., 2018).

Previous studies indicated that biofilms promote antibiotic resistance of many Staphylococcus strains. The antibiotic resistance of biofilm cells is up to 1,000-fold greater than the free-living planktonic bacterial cell (Ceri et al., 1999; Costerton et al., 1995; Larsen & Fiehn, 1996; Wentland et al., 1996). Since the MRSA strains used in this study are sensitive to vancomycin, we attempted to determine, in the presence of vancomycin (below the MIC concentration), whether or not the addition of stigmata maydis aqueous extracts would affect the survival of MRSA strains cultured under different conditions. The results showed that the stigmata maydis aqueous extracts cannot affect the vancomycin-sensitivity of MRSA bacteria grown in planktonic culture, but can significantly increase the sensitivity of MRSA bacteria grown in biofilm. Because the MRSA bacteria grown in biofilm have higher vancomycin resistance compared with that grown in planktonic culture, and stigmata maydis inhibits the biofilm formation of the MRSA bacteria, it indicates that the effect of stigmata maydis on vancomycin-sensitivity of MRSA bacteria grown in biofilm is through the inhibition of biofilm formation. These findings corroborated the previous study that reported that MRSA in planktonic growth was susceptible to vancomycin, however, the MIC of vancomycin for ica-positive MRSA tremendously increased (Chopra, Harjai & Chhibber, 2015).

Since stigmata maydis is an inexpensive and easily available Chinese medicine, it could be used as an ancillary aid to vancomycin treatment of MRSA, providing new clues for the prevention and control of bovine mastitis caused by MRSA strains.

Conclusions

The aqueous extracts of stigmata maydis inhibit the biofilm formation ability of MRSA strains isolated from dairy cows with mastitis by down-regulating the transcription of the ica operon. Moreover, the effect of stigmata maydis on vancomycin-sensitivity of MRSA in biofilm was through inhibiting the biofilm formation of the MRSA bacteria.

Supplemental Information

Supplemental Information 1 The supplemental files in Origin format.

Click here for additional data file.

Additional Information and Declarations

Competing Interests

Author Contributions

Data Availability

The authors declare that they have no competing interests.

Fei Shang performed the experiments, prepared figures and/or tables, authored or reviewed drafts of the paper.

Long Li performed the experiments, prepared figures and/or tables.

Lumin Yu analyzed the data, approved the final draft.

Jingtian Ni analyzed the data.

Xiaolin Chen contributed reagents/materials/analysis tools.

Ting Xue conceived and designed the experiments, contributed reagents/materials/analysis tools, authored or reviewed drafts of the paper, approved the final draft.

The following information was supplied regarding data availability:

The raw data are available in the Supplementary Files.

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
