# Peer review of "Effects of stigmata maydis on the methicillin resistant Staphylococus aureus biofilm formation"

_PeerJ, doi:10.7717/peerj.6461_

## Round 0.1 · original submission · Major Revisions

As you will see, your submission triggered quite contrasting opinions. We wish to give you the opportunity to respond to the major critiques. If you are ready to do the required work, please submit a revised version together with a detailed rebuttal where you explain how you have dealt with each critique. This rebuttal will be essential for us to make a final decision. Please, be aware that your revised version may undergo a new round of review by the same or by different reviewers. I cannot, therefore, make any commitment at this stage concerning a final acceptance of your paper.

·

Basic reporting

No comment

Experimental design

No comment

Validity of the findings

No comment

Additional comments

The authors have a good idea, but unfortunately, the introduction, the results and the discussion are complicated.

The introduction requires a careful review of the current literature and the authors should clearly indicate the problem as well as the objectives. please delete the text on line number 90-96, the text was a results.

In the results, as reported by the authors, that the aqueous extract of stigmata maydis can inhibit the formation of biofilms of MRSA strains and does not affect the growth curves of MRSA strains and has no antibacterial activity against MRSA strains, in this case how the aqueous extract could enhance the vancomycin susceptibility of the MRSA strains under biofilm-cultured conditions, and the plankton cells of MRSA strains has not been affected by the same of aqueous extract? The effects of the aqueous extract must be the same in either free planktonic cells or in biofilm formations, only the dose of antibacterial (if there is any) can present a significant difference in the formation of biofilms compared to free plankton cells.
In the next line, the text must delete, these are not results
Line 173-178
Line 190-195
Line 208-217

The discussion did not correspond to the objectives or the problem. Neither previous work nor real-time PCR results were well discussed. Most of the sentences were repeated from the results and methods. Finally, I think the authors should check the effects of Stigmata maydis on biofilm adhesion factors to MRSA

·

Basic reporting

no comment

Experimental design

no comment

Validity of the findings

no comment

Additional comments

Comments:
1. Keywords: I suggest adding stigmata maydis.
2. Lines 100-104: Please provide detailed description of MRSA strains SA2 and SA3. As a clinical isolated MRSA strain, which kind of antibiotic resistance dose it possess?
3. Line 115: “…diluted to a final concentration of OD600 0.05” change to: “diluted to a final optical density (600 nm) of 0.05”.
4. Line 120: Please provide specific calculation methods used in CFU assays.
5. Line 139: check the culture medium used here, TSB or TSB containing glucose?
6. Line 132, Materials Methods: Crystal violet is usually measured around 590 nm and not 560 nm. Please check.
7. The paper should be reviewed by an English native speaker reviewer. There are some sentences that are not clear.

---

## Round 0.2 · Major Revisions

As you can see, your revised version still triggered many remarks and criticisms by one of the reviewer who provided an annotated manuscript. I concur with many of these comments and suggestions for further improvement. So, please, provide a new rebuttal and revise further your text and/or explain why you disagree. i'm looking forward to hear from you in due course.

·

Basic reporting

please correct all highlights text in PDF

Experimental design

no comment

Validity of the findings

no comment

Additional comments

please correct all highlights text in PDF

·

Basic reporting

no comment

Experimental design

no comment

Validity of the findings

no comment

Additional comments

Thank you for responding to my original review comments and I do feel the manuscript now reads better. The paper is written within the Journal required style, It is suitable for publication in its present form.

---

## Round 0.3 · accepted · Accept

As you can see, your revision was successful and I guess you will agree that your paper is stronger.

# ·

Basic reporting

nil

Experimental design

nil

Validity of the findings

nil

Additional comments

please see the notes in annotated PDF